# Digital Transformation in Musculoskeletal Ultrasound: Acceptability of Blended Learning

**DOI:** 10.3390/diagnostics13203272

**Published:** 2023-10-20

**Authors:** Andreas Michael Weimer, Rainer Berthold, Christian Schamberger, Thomas Vieth, Gerd Balser, Svenja Berthold, Stephan Stein, Lukas Müller, Daniel Merkel, Florian Recker, Gerhard Schmidmaier, Maximilian Rink, Julian Künzel, Roman Kloeckner, Johannes Weimer

**Affiliations:** 1Clinic for Trauma and Reconstructive Surgery, University Clinic Heidelberg, 69118 Heidelberg, Germany; andreas.weimer@kkh-bergstrasse.de (A.M.W.);; 2Group Practice of Physicians Spilburg Wetzlar, Department of Orthopedics, 35578 Wetzlar, Germany; 3Rudolf Frey Learning Clinic, University Medical Centre of the Johannes Gutenberg University Mainz, 55131 Mainz, Germany; 4Department for Orthopaedics and Trauma Surgery, University Medical Centre Mannheim, 68167 Mannheim, Germany; 5Department of Diagnostic and Interventional Radiology, University Medical Centre of the Johannes Gutenberg University Mainz, 55131 Mainz, Germany; lukas.mueller@unimedizin-mainz.de; 6BIKUS—Brandenburg Institute for Clinical Ultrasound, Brandenburg Medical School Theodor Fontane (MHB), 16816 Neuruppin, Germany; 7Department of Obstetrics and Prenatal Medicine, University Hospital Bonn, 53127 Bonn, Germany; 8Department of Otorhinolaryngology, Head and Neck Surgery, University Hospital Regensburg, 93053 Regensburg, Germany; maximilian.rink@klinik.uni-regensburg.de (M.R.); julian.kuenzel@klinik.uni-regensburg.de (J.K.); 9Institute of Interventional Radiology, University Hospital Schleswig-Holstein—Campus Lübeck, 23538 Luebeck, Germany; roman.kloeckner@uksh.de

**Keywords:** ultrasound education, digitalization, digital transformation, blended learning, webinar, e-learning, ultrasound, teaching, musculoskeletal ultrasound

## Abstract

Background: ultrasound diagnostics have a broad spectrum of applications, including among diseases of the musculoskeletal system. Accordingly, it is important for the users to have a well-founded and up-to-date education in this dynamic examination method. The right balance between online and in-class teaching still needs to be explored in this context. Certifying institutions are currently testing digitally transformed teaching concepts to provide more evidence. Methods: this study compared two musculoskeletal ultrasound blended learning models. Model A was more traditional, with a focus on in-person teaching, while Model B was more digitally oriented with compulsory webinar. Both used e-learning for preparation. Participants completed evaluations using a seven-point Likert scale, later converted to a 0–1 scale. Digital teaching media (e-learning) were used for preparation in both courses. Results: the analysis included *n* = 41 evaluations for Model A and *n* = 30 for Model B. Model B received a better overall assessment (median: 0.73 vs. 0.69, *p* = 0.05). Model B also excelled in “course preparation” (*p* = 0.02), “webinar quality” (*p* = 0.04), and “course concept” (*p* = 0.04). The “gain of competence” (*p* = 0.82), “learning materials” (*p* = 0.30), and “tutor quality” (*p* = 0.28) showed no significant differences. Conclusion: participants favorably assessed blended learning in ultrasound teaching. Certifying institutions should consider accrediting models that combine digital methods (e.g., internet lectures/webinars) and materials (e.g., e-learning) with hands-on ultrasound training. Further research is needed to validate these subjective findings for a stronger evidential basis.

## 1. Introduction

### 1.1. Status Quo: Teaching of Ultrasound to Physicians in Germany

Ultrasound is an established standard examination in many specialties. For imaging of the musculoskeletal system, ultrasound can be—along with X-ray, CT, and MRI—decisive for differential diagnostics [1,2,3]. Accordingly, it is important for the users to have a well-founded and up-to-date education in this dynamic examination method. In order to optimize education, innovative teaching possibilities are currently being discussed in the framework of digital transformation [4,5,6,7]. Yet, there are still no standard guidelines for such digitalized educational concepts and their certification. Written standards for undergraduate and postgraduate medical education are provided in the continuing education catalogs of the German Medical Association and the guidelines of the Federal Association of Statutory Health Insurance Physicians [8,9]. In addition to those associations, specialist societies such as the German Society for Ultrasound in Medicine can also certify competence in ultrasound [10]. The educational curricula certified by these societies usually consist of basic courses, advanced courses, and certification courses with defined theory and practice units [11,12,13].

### 1.2. Teaching Materials, Teaching Methods, and Course Formats

In addition to analogue teaching materials (mainly books or collections of cases), digital teaching materials such as e-learning are increasingly being used for preparation, accompanying the course, and later review [14]. “E-learning” refers to digital technologies that are used for transmission of knowledge during the entire learning process [15,16]. The so-called “blended learning” concept has firmly established itself as a teaching method in medical education. Applications in ultrasound training have already been described [17,18,19]. This teaching concept combines in-person activities with digital teaching formats [20]. Increasingly, webinars [6,21,22,23,24] are being used to communicate theoretical contents in a location-independent, standardized, and cost-efficient way [25,26,27,28]. Most virtual platforms offer the possibility to share videos or slideshow presentations with other webinar participants or provide them on demand.

### 1.3. Research Problem and Aim

In the context of musculoskeletal ultrasound education, a crucial observation surfaces: failing to acknowledge the significance of employing a systematic approach when covering various joints. A systematic approach in education is vital for a thorough exploration of diverse joints, considering their intricate anatomical features and potential pathologies. Such an approach is indispensable for accurate diagnoses and the subsequent implementation of appropriate medical interventions [29,30]. The heavy workload during the clinical routine and consequent shortage of available time for other activities force education to the back seat. This is especially true for resource-demanding activities like ultrasound training. Offering courses and postgraduate education opportunities that are characterized by factors such as effectiveness, time saving, resource-consciousness, and sustainability could help optimize educational pathways. Awareness of the digital transformation has also grown due to the constraints regarding face-to-face activities during the COVID-19 pandemic [4,7,31]. Certifying institutions such as the Federal Association of Statutory Health Insurance Physicians have called for “more flexibility for the participation in ultrasound courses” and have scheduled a three-year test phase [7]. During that period, evidence for or against online/digitally supported ultrasound courses should be gathered. Until now, only a few studies exist that have compared traditional ultrasound courses with digitalized ultrasound courses [32]. We were unable to find any studies focused on musculoskeletal ultrasound (MSUS). Thus, the present study aims to contribute to filling this gap by studying how the participants evaluate the increased use of digital teaching methods and materials within MSUS courses certified by the German Society for Ultrasound in Medicine (DEGUM). Hence, the primary outcome of the study is the acceptance of digital versus traditional ultrasound education in MSUS. Secondary outcomes distilled from the evaluation describe the digital aspects of the blended learning components of the courses. Our primary hypothesis is that both formats will be equally accepted by the participants. Our secondary hypothesis is that participants appreciate the digital aspects of blended learning. By analyzing subjective evaluations of the participants, this study will provide initial experience regarding the acceptance of digitalized ultrasound courses in modern ultrasound education.

## 2. Methods

### 2.1. Study Design, Study Process, and Overview of Teaching Concepts

This prospective observational study was carried out from 2021 to 2022. We evaluated the satisfaction and, thus, the inferred acceptance of a digitally transformed course model in comparison to the classically delivered education model used until now in a DEGUM-certified musculoskeletal ultrasound course. First, ultrasound education models for MSUS were analyzed based on the current literature and training curricula of professional societies. Next, a traditional course, Model A (mainly face-to-face teaching), was transformed into a course with more webinar and digital media (Model B). This was carried out in compliance with the standards and guidelines of the German Society for Ultrasound in Medicine and the Federal Association of Statutory Health Insurance Physicians [7,9,13]. Though both course models contained blended learning elements, Model B contained considerably more digital media and webinar time [18]. The primary endpoint of the study was the assessment of the teaching models by means of a digital evaluation questionnaire. The inclusion criteria were the complete participation in the course and filling out the evaluation questionnaire completely. The study process, including both models, is presented in Figure 1.

### 2.2. Description of Course Model A

Model A offers a predominantly in-person musculoskeletal-focused basic ultrasound course spanning three days, with a balanced 50:50 ratio of theory to hands-on practice. In preparation for this course, participants had the option to voluntarily attend a 60 min informational webinar covering course details, preparatory guidance, a brief overview of the topics, and fundamental ultrasound principles. Additionally, participants were granted early access to e-learning materials, including recorded webinars and examination videos, for the following musculoskeletal areas: “shoulder”, “elbow”, “hand”, “hip”, “knee”, and “foot” (accessible via a video platform). These e-learning resources required approximately 60 min for completion.

During the in-person phase of the course, the curriculum covered examination techniques, normal findings, and common pathologies through systematic live lectures lasting approximately 30–60 min per joint or muscle region. These lectures were presented using slideshow presentations and illustrated with case studies showcasing various pathologies. To aid in their learning, all participants were provided with lecture notes summarizing the key normal findings for the sonographic sections featured in the course. The practical training on the ultrasound devices took place with a 5:1 participant to tutor ratio. An anatomy ultrasound app (Atlas der Humananatomie 2021, Version 2021.2.24, © Visible Body) [33] was also used in addition to the ultrasound training. All lecturers and instructors were physicians certified by DEGUM. In addition to providing participants with workbooks and e-learning materials, the course distributed a sonography poster and a printed sheet of findings for participants to use for later study.

### 2.3. Description of Educational Model B

Model B was developed on the basis of Model A but focused increasingly on digitalization and blended learning. The course was conducted over three days with the same relation of theory to practice. The same tutors and course leader as in Model A were involved. Class sizes and learner to tutor ratio (5:1) were identical. Pathologies were taught based on clinical cases as well.

In contrast to Model A, Model B featured a course structure consisting of one digital and two in-person course days. The digital portion included a mandatory 180 min webinar (comprising 4 instructional units) that was recorded, and participants’ attendance was verified through chat registration. Additionally, participants were granted access to 4 extra instructional units of voluntary digital preparation via e-learning. Unlike Model A, Model B’s e-learning included narrated stimulus lectures on various joint regions, accessible on a video platform, in addition to the video links.

During the in-person phase, the teaching approach mirrored that of Model A, with shorter stimulus lectures (approximately 15–20 min per joint/muscle region). The use of lecture notes and an anatomy app remained consistent with Model A. The distribution of the sonography poster, findings sheet, and course evaluation process also followed the same protocol as in Model A. For further study, participants had access to the workbook, e-learning materials, and recorded webinar videos.

### 2.4. Questionnaire

A digital questionnaire was developed for the evaluation of both course models. It comprised the following sections: “baseline”, “evaluation of the course”, “evaluation of the webinar”, “course preparation”, “evaluation of the teaching material”, “self-evaluation of competency”, and “evaluation of the tutors”, with several subitems per section (Table 1), and mainly used a seven-point Likert scale (one = complete agreement, seven = absolutely no agreement) as well as dichotomous questions (“yes”/“no”).

### 2.5. Statistical Analysis

The evaluations were conducted digitally through an online questionnaire tool and were exported to an Excel spreadsheet. Next, data were imported into the statistics program R studio with R 4.0.3 (R Foundation for Statistical Computing; A Language and Environment for Statistical Computing). After import, data cleaning was performed. A main scale score (for each topic area shown in the table) was made from the average of the subscale scores. Furthermore, an overall score was calculated from the average of the main scale scores. The internal consistency of the scales was tested and ensured by calculating the reliability according to Cronbach’s alpha. In order to make the subscales fit together, they were transformed to a zero to one range, whereby one corresponds to the maximum (formerly seven on the Likert scale) and zero to the minimum (formerly one on the Likert scale). Then, descriptive, exploratory, and inferential statistics were performed. The interval scales were mathematically assessed for normal distribution by means of Shapiro–Wilk tests. For normally distributed scales, the Welch two-sample *t*-test was used. For the scales that were not normally distributed, the nonparametric Wilcoxon–Mann–Whitney test was used. For the comparison of subscales within each course model, a post hoc test was carried out. *p*-values < 0.05 were considered statistically significant.

## 3. Results

### 3.1. Data Description

The reliability tests, according to Cronbach’s alpha, show that the internal consistency of the scales, in a range of 0.82–0.95, did not vary considerably.

### 3.2. Study Population

The analysis included evaluations collected from 71 participants: *n* = 41 for course Model A, and *n* = 30 for course Model B. The demographic data, as well as the results for the course preparation and webinar participation, are presented in Table 2. The participants of Model B claimed to have ultrasound experience more often than did the participants of Model A, and they used the e-learning more often to prepare for the course (100% vs. 63%). In both models, e-learning and online videos were used more often than textbooks (A: 44%, B: 40%) or other online resources (A: 49%, B: 33%). Altogether, 49% of Model A participants and 93% of Model B participants participated in the webinar.

### 3.3. Survey Results

The evaluation results overall and per topic are presented in Figure 2. Model B achieved slightly higher ratings than Model A (median: 0.73 vs. 0.69, *p* = 0.05). There was no significant difference for “gain of competence” (mean of 0.23 in both groups, *p* = 0.82), “teaching materials” (mean of 0.74 in A vs. 0.77 in B, *p* = 0.30), or the “tutors” (median 0.90 in A vs. 0.92 in B, *p* = 0.28). In Model B, there was a significantly higher rating of the “course preparation” (A: median 0.33, IQR [0.00–0.33] vs. B: median 0.33, IQR [0.33–0.67]; *p* = 0.02), the “evaluation of the webinar” (0.87 vs. 0.68, *p* = 0.04), and the “evaluation of the course” (0.82 vs. 0.69, *p* < 0.01), compared to Model A.

The evaluation results of the subitems of the main scales are visualized in box-and-whisker plots in Figure 3 and Appendix A. The contents of the webinar (median of 0.75 in A vs. 0.88 in B), the course concept (median of 0.5 in A vs. 1.0 in B, *p* = 0.04), the time commitment (median of 0.5 in A vs. 1.0 in B, *p* < 0.01), and the relation between theory and practice (median of 0.75 in A vs. 1.0 in B, *p* = 0.03) were on the upper half of the scale range for both models but were often significantly higher for Model B. This was also true for the evaluation of the ultrasound app used (A: median 0.8, IQR [0.60–0.80] vs. B: median 0.8, IQR [0.80–1.00]; *p* = 0.02). The e-learning (median of 0.75 in A vs. 0.75 in B, *p* = 0.09), the case examples in general (median of 0.75 in A vs. 1.0 in B, *p* = 0.12), and the slideshow presentation (median of 0.67 in A vs. 1.0 in B, *p* = 0.33) were also in the same upper range of the scales, without significant differences. The more specific evaluation of the more comprehensive webinar of course Model B is also presented in Figure 3. Here, the time commitment of the webinar (median [IQR]: 0.75 [0.5–1]), the recorded webinar videos (median [IQR]: 1.0 [0.88–1.0]), and the webinar lectures (median [IQR]: 1.0 [0.5–1.0]) were especially highly rated, yet without significant differences between these items (*p* > 0.05).

A CrossCorr Heatmap of the items is presented in Appendix A. This heatmap shows that, overall, a positive evaluation of the webinar goes along with a positive evaluation of the course (gradient 0.60, *p* = 0.01). But the entire course is positively evaluated when the teaching materials—also digital ones—are viewed positively (gradient 0.74, *p* < 0.01).

## 4. Discussion

This study aimed to prospectively assess the satisfaction and acceptance of blended learning approaches in two musculoskeletal ultrasound course models; first, a traditional ultrasound course (A: mainly face-to-face learning) and, second, a digitalized version of the same courses (B: more digital formats, e.g., webinar). In summary, digitalization processes within certified sonography courses were fully accepted and supported by the participants. The better evaluation of course Model B, in particular, regarding “concepts”, “webinar”, and “digital preparation”, favors the integration of more digital teaching formats in future sonography education and the further adaption of current educational concepts towards these digital formats. Certifying institutions should also take this into account in future accreditation of teaching concepts.

### 4.1. Discussion of “Blended Learning” Formats

The positive evaluation results of our study should be taken as an opportunity to optimize ultrasound course concepts in the future. Therefore, in regards to the instructional location, the role of the course leader (background or foreground), and the communication (analogue or digital), “classic in-person” courses with the use of offline teaching materials (e.g., slideshow presentation) should be transformed in the future into “blended learning” course (in-person and online learning) [14,34]. Such concepts offer a wide range of opportunities and possibilities. Reduced time and personnel requirements free up working time of physicians. Reduced travel saves money and is more eco-friendly, an aspect becoming increasingly important in the healthcare sector. Digital concepts allow physicians to continue training despite restrictions imposed on face-to-face teaching [32,35,36,37]. While this trend can already be increasingly observed in the teaching of ultrasound to students, it is still lagging in postgraduate education of physicians [4,7,38,39]. Only a few studies compared traditional education with digitally transformed education. In the area of point-of-care ultrasound, the acceptance and the equivalent use of a hybrid teaching model, in comparison to a traditional approach, has already been shown [32]. Our data for MSUS also support these results. Blended learning activities, as in Model B, should be more intensively integrated into medical education and postgraduate medical education [4,19,32] and more widely accepted by the certifying institutions [4,7]. Furthermore, the number of recent scientific publications on blended learning indicates its increasing importance, especially regarding a sustainable and efficient postgraduate education of ultrasound diagnostics [6,17,18,20,32,40]. This is also supported by the increased use of ultrasound applications on mobile devices, one of which was positively evaluated in this study [41].

The agreement recorded in our study for the item “case-based learning” can also be taken as a reason to digitally transform this to a greater extent in the future [42,43]. Theoretical contents may also be taught in “massive open online courses” (MOOCs), to address a larger number of participants [44]. This is an opportunity for professional societies to provide up-to-date teaching materials to all members. Furthermore, the significance of employing a systematic approach to cover various joints in musculoskeletal ultrasound education is essential and should be imparted during every MSUS course [29,30].

### 4.2. Discussion of the (Digital) Course Preparation

Figure 2 show a preference to use digital formats for preparation, especially on-demand online resources. Various digital teaching media, such as websites [45], online atlases [46], or video platforms, may be used. Not all readily available teaching materials have undergone quality control by experts and a certain competency of the users must be assumed. Digitalization material (e.g., transformation of an analogue textbook into an e-learning format) should also be considered from an economic viewpoint [41]. If necessary, resources and materials should be combined to develop teaching materials’ cost efficiently and to make them accessible to a broad spectrum of users. The relevant professional societies should be more involved in this process and take initiative [47].

### 4.3. Discussion of Webinar versus In-Person Teaching

In addition to e-learning, the webinar is a central tool of the digital transformation of the teaching model in this study. We strongly pursued the goal of delivering more teaching content digitally through webinars, not least because of the COVID-19 pandemic, in medical education and also ultrasound training [6,26,48]. In this study, the use of webinars was fully accepted and supported by the participants, as shown by the positive evaluations.

Nonetheless, the digital webinars have advantages and disadvantages [22,23,26,27,28,48]. A positive relationship between webinar teaching and an improvement of knowledge, behavior, and abilities was demonstrated in a meta-analysis and systematic literature review [22], which is also confirmed in part by the evaluation data of the present study (subjective growth of competence). Positive aspects of webinar teaching are the flexibility in time and place, cost-efficiency (for presenters and participants), and the digital review possibilities via recordings [26]. These recordings are termed “on-demand webinars” and are well suited for topics with a timeless character (“evergreen content”), such as the basic sono-anatomical information, the probe positioning, or storing. Likewise, webinars offer the possibility of bringing together “experts” digitally, independent of time and place—asynchronously through recordings—to lecture about specialist topics. Leading professional societies are also currently following this trend, offering special topical webinars with expert panels and providing access to archived recordings [49]. However, possible technical problems must be considered, especially in regards to the necessity of a stable and fast internet connection and sufficient server capacity as well as the problems of pure “virtual communication” (limited group dynamics and, possibly, the lack of informal exchanges among the participants), which requires a special competence of the participants [28]. The lattermost challenge mentioned can be managed by using “breakout rooms”, which enable exchanges of the webinar participants in small group formats. Altogether, training the webinar teaching staff also represents an important aspect in the delivery and quality assurance and should, in the future, be taken into consideration in “train the trainer” courses [50,51]. Whether the teaching via webinar or e-learning was adequate, standardized examination formats, such as multiple-choice tests, may be administered [52]. The certifying institutions should be involved in the development process. Course models that contain, where applicable, several smaller “webinar slots” could lead to a more efficient preparatory phase in order to build up relevant sonographic competences for practical application during the in-person phase and to enable a more sustainable growth of knowledge.

### 4.4. Limitations

It is important to note that this study was conducted at a single center, which may limit the generalizability of our findings, limiting the extent to which educators in MSUS can rely on these recommendations. We cannot completely rule out that the variations in the duration of the e-learning in the preparation phase influences the study’s outcomes, especially regarding the participant’s attitude towards the digital course aspects. Although the internal consistency of the scales was confirmed by reliability tests according to Cronbach’s alpha, a complete validation of the questionnaire did not take place. Due to the variable performance level (prior clinical experience, previously acquired knowledge in the use of ultrasound diagnostics), a standardization or quantification of the subjective teaching success was possible only to a limited degree. Further, participants’ evaluations and feedback were based on self-reporting, which can introduce biases. Participants may provide responses that align with their personal preferences or perceptions, and these may not always align with the objective effectiveness of the course models. Our study’s results are dependent on participants’ willingness to complete the survey. Response bias may exist if participants with particularly strong opinions, either positive or negative, were more inclined to respond, potentially skewing the overall feedback. It is important to note that this study was conducted at a single center, which may limit the generalizability of our findings. Different educational settings or institutions with varying resources and teaching approaches may yield different results. The study acknowledged the impact of the COVID-19 pandemic on the increased use of webinars. This external factor may have influenced participants’ preferences and perceptions during the study period, potentially affecting their feedback. Finally, it is essential to recognize that our study focused on MSUS. The findings may not directly apply to other medical specialties or educational domains.

### 4.5. Conclusions

Considering the mentioned limitations, our study provides valuable insights into the satisfaction and acceptance of different course models in MSUS. According to the data collected in this study, digital teaching concepts, including methods and media, were evaluated positively by the participants in the context of postgraduate medical education of ultrasound of the musculoskeletal system. Participants were more satisfied with the combined use of a webinar and digital materials such as e-learning in a digitally transformed course model. These results show promising ways to establish more blended learning-based courses in ultrasound education in the future, and those should be supported more by the certifying institutions. To confirm the subjective results collected in this study, further studies should focus on the acquisition of objective–theoretical and practical competences.

## Figures and Tables

**Figure 1 diagnostics-13-03272-f001:**
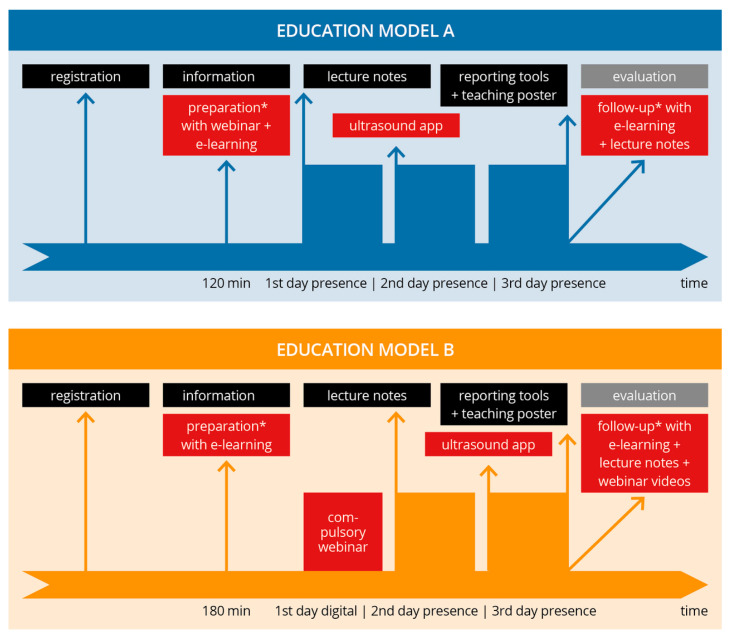
Study process and overview of the educational models. Flowchart providing the schedule of both course models (A: traditional; B: blended learning). The digital elements of both models are shown in red. In both models, the same evaluation (gray box) was performed, representing the endpoints of the study. * voluntary part of the course.

**Figure 2 diagnostics-13-03272-f002:**
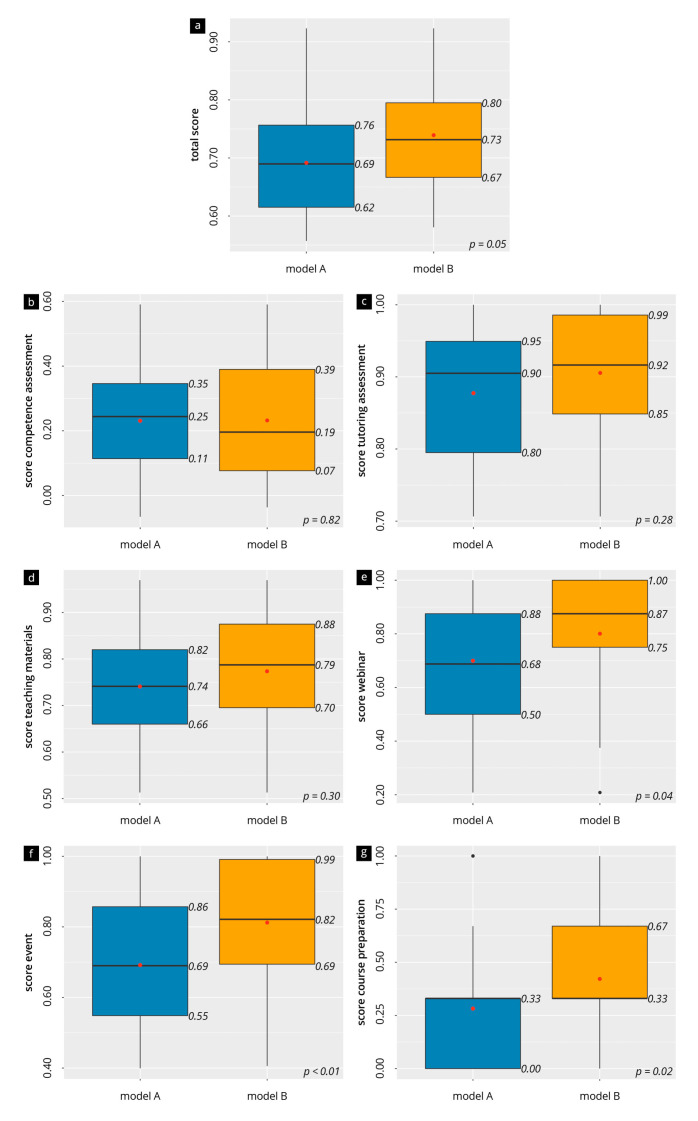
Evaluation results of the averaged items of the main topics (**a**–**g**). The box-and-whisker plots visualize the scores (high numbers = high agreement); the median (black line) and mean (red dot) are also shown.

**Figure 3 diagnostics-13-03272-f003:**
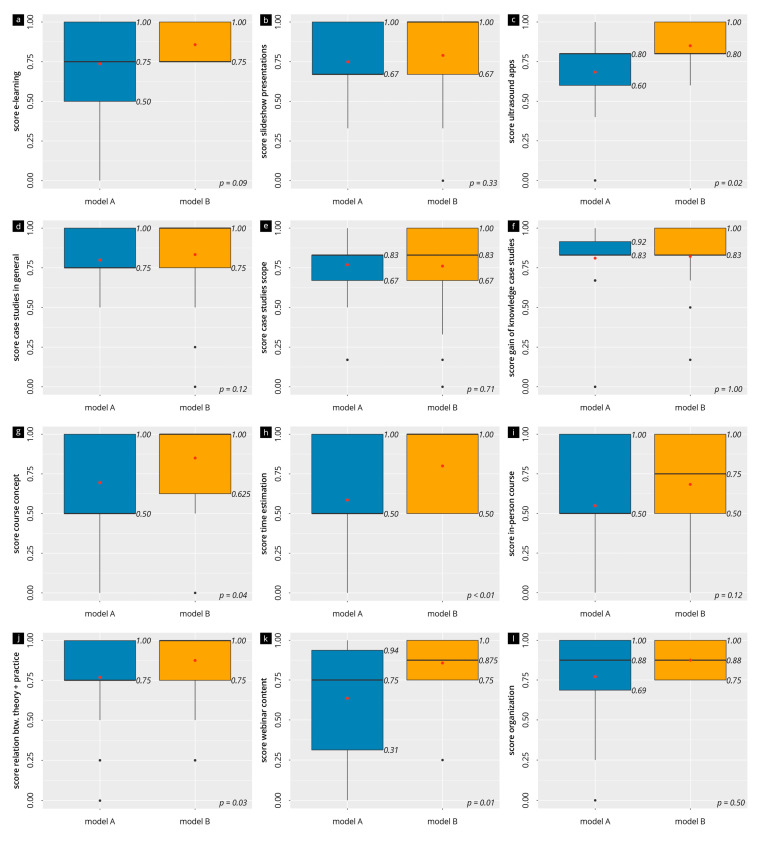
Evaluation results of the individual items. The box-and-whisker plot visualization of the ratings of the teaching materials (panels (**a**–**f**)), the evaluation of the course (panels (**g**–**j**)), and the evaluation of the webinar (panels (**k**,**l**)). The black line in the middle of the box represents the median, and the red dot represents the mean.

**Table 1 diagnostics-13-03272-t001:** Overview of the items and subitems of the evaluation questionnaire. Items marked with a “*” were only asked for regarding course Model B.

Topic Area	Subitems	
Baseline	age sex	position sonography experience
Course	fulfillment of expectations in-person course organization course concept	achievement of the learning goals time estimation relation between theory and practice
Webinar	participation in webinar organization contents webinar in general *	time estimation of the webinar * webinar lectures * teachers of the webinar * videos after the webinar *
Course Preparation	amount of time e-learning (video links)	textbooks online resources
Teaching Materials	teaching materials in general e-learning lecture notes ultrasound app	slideshow presentation case examples amount of case examples gain of knowledge through the case examples
Competency Self-Evaluation	specialist knowledge operation of the device ultrasound probe handling spatial orientation sono-anatomical mapping muscle, joint, and soft tissue representation muscle and joint evaluation	patient guidance shoulder elbow hand and fingers hips and thighs knee joint ankle and foot orientation levels/standard planes
Evaluation of the Tutors	overall specialist knowledge operation of the device ultrasound probe handling spatial orientation sono-anatomical mapping muscle, joint, and soft tissue representation muscle and joint evaluation	patient guidance general demeanor communication with the group use and handling of the learning materials presentation of the topics handling of questions

**Table 2 diagnostics-13-03272-t002:** Demographic data and results for the course preparation and participation in the webinar.

	Model A (*n* = 41)	Model B (*n* = 30)
Age (mean [SD])	39.0 [8.3]	37.4 [8.0]
Sex		
female (*n* [%])	13 [32]	15 [50]
male (*n* [%])	28 [68]	15 [50]
Position		
medical student (*n* [%])	0 [0]	2 [6.7]
medical resident (*n* [%])	18 [44.8]	11 [36.7]
physician (*n* [%])	16 [39.7]	14 [46.7]
senior physician (*n* [%])	5 [12.1]	2 [6.7]
department chief (*n* [%])	1 [3.4]	1 [3.3]
Ultrasound Experience		
beginner (*n* [%])	30 [73.4]	23 [76.6]
advanced (*n* [%])	11 [27.6]	7 [23.3]
Course Preparation		
textbooks		
yes (*n* [%])	19 [46.3]	12 [40.0]
no (*n* [%])	22 [53.7]	18 [60.0]
e-learning (+videos)		
yes (*n* [%])	26 [63.4]	30 [100.0]
no (*n* [%])	15 [36.6]	0 [0.0]
online offerings		
yes (*n* [%])	20 [48.8]	10 [33.3]
no (*n* [%])	21 [51.2]	20 [66.7]
webinar participation		
yes (*n* [%])	21 [51.2]	28 [93.3]
no (*n* [%])	20 [48.8]	2 [6.7]

## Data Availability

Data cannot be shared publicly because of institutional and national data policy restrictions imposed by the Ethics committee since the data potentially contain information identifying study participants. Data are available upon request (contact via weimer@uni-mainz.de) for researchers who meet the criteria for access to confidential data (please provide the manuscript title with your enquiry).

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
