# Peer review of "Digital Transformation in Musculoskeletal Ultrasound: Acceptability of Blended Learning"

_diagnostics, 2023, doi:10.3390/diagnostics13203272_

Round 1
Reviewer 1 Report
This article reports on an evaluation of two approaches to teaching musculoskeletal ultrasound: a traditional ‘hands on’ approach (Model A) versus a ‘blended’ approach combining hands on training with more self-directed online digital content (Model B). At first glance, the description of the two course plans looks very reasonable, with Model B still providing two thirds of the ‘hands on’ time compared to Model A. If confirmed, the results of this study will encourage more use of hybrid/blended models such as this, potentially increasing access for busy clinicians and reducing cost. Unfortunately, there are some deficiencies in the study design and methodology that limit the extent to which educators in MSUS can rely on their recommendations.
Minor editing required
Author Response
Thank you very much for giving feedback on the questions which we fully addressed in our p2p. Please find attached our comments and corrections more detailed in the pdf file.

Reviewer 2 Report
This study aims to present initial findings on the acceptance of digitally transformed ultrasound courses in contemporary ultrasound education. Overall, I find the article intriguing, but I have several comments that I hope the authors will take into consideration.
Firstly, the authors did not address the significance of employing a systematic approach to cover various joints in musculoskeletal ultrasound education. I recommend referencing the following articles to strengthen this aspect:
- https://bmcmededuc.biomedcentral.com/articles/10.1186/s12909-019-1769-6
- https://www.sciencedirect.com/science/article/abs/pii/S0301562923000595
Secondly, it is essential that the study's objectives are explicitly stated at the end of the introduction section.
Thirdly, it would be beneficial if the authors could articulate their hypothesis regarding the expected outcomes of the study.
Fourthly, please provide information regarding the Institutional Review Board (IRB) approval for this study to ensure its ethical compliance.
Fifthly, I noticed that there might be variations in the length of education between the two models being compared. Could these differences potentially influence the study's outcomes? It would be helpful if the authors could address this potential factor.
Lastly, has the questionnaire used in this study undergone validation to establish its reliability and effectiveness as a research tool? This information would bolster the methodological rigor of the study.
Author Response

(The authors gave the same response as above.)
